# Promoter Identification and Transcriptional Regulation of the Goose *AMH* Gene

**DOI:** 10.3390/ani9100816

**Published:** 2019-10-16

**Authors:** Shuang Yang, Yan Deng, Da Chen, Shenqiang Hu, Yingying Zhang, Huilan Huang, Jiwei Hu, Liang Li, Hua He, Jiwen Wang

**Affiliations:** Farm Animal Genetic Resources Exploration and Innovation Key Laboratory of Sichuan Province, College of Animal Science and Technology, Sichuan Agricultural University, Chengdu 611130, China; ys810997989@163.com (S.Y.); dengyan3691@163.com (Y.D.); chd00322@163.com (D.C.); shenqianghu@gmail.com (S.H.); zyy11email@163.com (Y.Z.); huilan_huang123@163.com (H.H.); hujiwei1990@126.com (J.H.); Liliang@sicau.edu.cn (L.L.); hehua023@126.com (H.H.)

**Keywords:** Goose, AMH, granulosa cells, promoter analysis, transcriptional regulation

## Abstract

**Simple Summary:**

Anti-Müllerian hormone (AMH) plays a vital role in the development of follicles. We found that the cloning nucleotide sequence of *AMH* was high homology in geese with other species. Several regulatory elements were identified and transcriptional factors were predicted in the *AMH* promoter sequence. Through a double-luciferase reporter assay, potential regulatory relationship spanning from −637 to −87 bp were identified. In addition, the mRNA expression of *AMH* gradually decreased during the development of follicles in geese. In the Chinese hamster ovary (CHO) cell line, the luciferase activity significantly increased by co-expression of *AMH* and *GATA-4*. However, when the binding sites of *GATA-4* to the promoter of *AMH* were mutated, the luciferase activity significantly decreased. These results indicated that the transcription of *AMH* was activated by *GATA-4* to inhibit the development of follicles in geese.

**Abstract:**

Anti-Müllerian hormone (AMH) is recognized as a reliable marker of ovarian reserve. However, the regulatory mechanism of goose *AMH* gene remains poorly understood. In the present study, both the full-length coding sequence (CDS) and promoter sequence of goose *AMH* have been cloned. Its CDS consisted of 2013 nucleotides encoding 670 amino acids and the amino acid sequence contained two structural domain: AMH-N and transforming growth factor beta (TGF-β) domain. The obtained promoter sequence spanned from the −2386 bp to its transcription start site (ATG). Core promoter regions and regulatory elements were identified as well as transcription factors were predicted in its promoter sequence. The luciferase activity was the highest spanning from the −331 to −1 bp by constructing deletion promoter reporter vectors. In CHO cells, the luciferase activity significantly increased by co-expression of *AMH* and GATA binding protein 4 (*GATA-4*), while that significantly decreased by mutating the binding sites of *GATA-4* located in the −778 and −1477 bp. Results from quantitative real-time polymerase chain reaction (qPCR) indicated that levels of *AMH* mRNA in geese granulosa layers decreased gradually with the increasing follicular diameter. Taken together, it could be concluded that the transcriptional activity of *AMH* was activated by *GATA-4* to inhibit the development of small follicles in goose.

## 1. Introduction

Anti-Müllerian hormone (AMH) is known as a member of the transforming growth factor-β (TGF-β) superfamily and exerts its biological actions by interacting with its specific type I and II receptors (AMHR I and II) [1]. With the sequence availability of *AMH* and its receptor across a range of species, accumulating evidence has shown that *AMH* plays a key role in the control of mammalian and avian gonadal functions, such as spermatogenesis and ovarian follicle development [2,3,4,5]. In mammals, *AMH* negatively regulated the recruitment of primordial follicles to avoid their excessive consumption during the reproductive cycle [6,7]. In mouse models, knockout of *AMH* resulted in a significant increase in the number of all growing follicles compared to the control group [2]. It was further demonstrated that *AMH* inhibited the growth of follicles by decreasing their sensitivity to follicle-stimulating hormone (FSH) [8,9,10]. In human granulosa cells (GCs) cultured in vitro, the mRNA expression of follicle-stimulating hormone receptor (*FSHR*) decreased after treatment with AMH [11]. These results suggested that *AMH* might inhibit the growth of follicles via the FSH/FSHR system in mammals. In avian, the mRNA expression pattern of *AMH* in GCs during follicle development was similar to that observed in mammals, with higher levels in small follicles [12]. In hens, the expression level of *AMH* mRNA significantly declined with the increasing follicular diameter [13]. In cultured chicken GCs from the 6 mm 8 mm follicles, *AMH* suppressed the FSHR signaling to prevent premature GCs differentiation, however, FSH signaling suppressed *AMH* expression to initiate GCs differentiation [14]. These results indicated that *AMH* played a key role during the development of small follicles in avian. The follicles of the avian ovary were arranged in a strict hierarchy, which were generally classified into the pre-hierarchical and hierarchical follicles [15]. The unique developmental characteristic of avian follicle contribute to define the causative factors which result in follicle atresia during the development of follicles in avian [16]. However, the sequence information and expression pattern of *AMH* in geese ovarian follicles still remain unknown.

Promoter is usually located in the 5′ proximal region of one gene and plays a pivotal role in gene expression. The promoter of *AMH* has been cloned in humans and mice [17,18]. At present, several potential transcription factors have been identified to be involved in the regulation of *AMH* gene expression. For example, steroidogenic factor 1 (SF-1) directly activated the transcription of *AMH* by binding to its proximal promoter in both embryonic and postnatal gonads [19]. In GCs of polycystic ovary syndrome (PCOS), FSH might inhibit the excessive secretion of AMH by suppressing its promoter activity [20]. Besides, the coordinated interplay between these two factors has also been found to regulate *AMH* expression. Barbara et al. showed that the *SRY*-box transcription factor 9 (SOX9), as an interaction partner of SF-1, could be related to the Sertoli cell-specific expression of *AMH* during embryogenesis [21]. In mouse GCs, Friend of GATA family member 2 (FOG2) was able to repress the effect of GATA-4 on *AMH* transactivation [22]. In addition, forkhead box protein L2 (FOXL2) was essential for SF-1-induced *AMH* regulation via protein-protein interactions during the development of human ovarian follicle [23]. Although some studies have partially revealed the mechanisms regulating *AMH* transcription in a number of species, little is known about such mechanisms in geese.

Therefore, the aim of the present study was to amplify the complete coding sequence (CDS) and the 5′-flanking region of goose *AMH* gene, to determine the relative levels of *AMH* mRNA in GCs during geese follicular development and to reveal the regulatory mechanisms underlying *AMH* transcription. These results will help broaden our understanding of the role of *AMH* in the goose ovary as well as its transcriptional mechanism during follicle development.

## 2. Materials and Methods

### 2.1. Animals and Sample Collection

The healthy female maternal line of Tianfu meat geese (Anser cygnoide) were approved by the waterfowl breeding of Sichuan Agricultural University. The name of the ethics committee is the Institutional Animal Care and Use Committee (IACUC) of Sichuan Agricultural University, the number/ID of the approval is DKY-B20141401.The geese had free access to water and food with natural lighting and temperature conditions. Geese aged 34–36 weeks during a regular period of egg laying were selected. Three geese were used for determining the relative abundance of *AMH* mRNA and six geese were used for collection of GCs for in vitro culture. Different classified follicles were collected at 7–9 h before oviposition. The granulosa layers from different diameters (4–6 mm, 6–8 mm, 8–10 mm) and different size (the fifth, fourth, third, second and the first larger follicles (F5 ~ F1)) follicles were separated as the methods of Gilbert and Deng [24,25]. All tissues were washed with ice-cold sterile phosphate buffered saline (PBS, pH 7.4. Solarbio, Beijing, China) and immediately frozen in liquid nitrogen until RNA extraction. All procedures were approved by the Institutional Animal Care and Use Committee (IACUC) of Sichuan Agricultural University (Permit No. DKY-B20141401).

### 2.2. Isolation of the Geese AMH CDS and Promoter

The predicted sequence of *AMH* gene (GenBank accession No. XM_013196337.1) and the goose genomic DNA sequence (GenBank accession No. NW_013185857.1) were obtained from NCBI (https://www.ncbi.nlm.nih.gov/gene/?term=AMH+Anser). The primers used for amplifying both the coding sequence and the 5′-flanking region of *AMH* were designed by the Primer Premier 5.0. The CDS sequence and 5′-flanking sequence were amplified by segmenting amplification and spliced using the SeqMan Pro. The fragments were overlapping with each other. Primers used for cloning CDS were listed in Appendix A and used for cloning 5’-flanking region were listed in Appendix A. Total RNA was isolated from granulosa layers of different diameters or sized follicles using TRIzol (Takara, Dalian, China). Complementary DNA (cDNA) was synthesized from 1 µg of total RNA using a cDNA synthesis kit (Takara, Dalian, China) following the manufacturer’s instruction. Genomic DNA was extracted using the granulosa layers of goose follicles via DNA Extraction Kit (Tiangen, Beijing China) following the manufacturer’s instructions. The cDNA and DNA from granulosa layers as templates, the polymerase chain reactions (PCRs) were performed with primer-specific annealing temperature for 30 s, 72 °C for 1 kb/min. Target fragments were ligated into the pMD19-T vector (Takara, Dalian, China) and were then transformed into *E. coli* DH5α competent cells. Positive clones that contained the expected-size fragments were screened by using colony PCR and were then sequenced by using the Qinke procedure (Chengdu, China).

### 2.3. Bioinformatics Analysis

Accuracy of cloning sequences was detected by the BlASTn program (https://blast.ncbi.nlm.nih.gov/Blast.cgi) and 5′-flanking sequence of goose *AMH* gene was confirmed on genomic DNA of geese (https://www.ncbi.nlm.nih.gov/assembly/GCF_000971095.1/). Homology analyses of nucleotide and amino acid sequences were performed using the DNAMAN software. Multiple alignments of *AMH* gene amino acid sequence were performed with the ExPASy ProtParam System (http://espript.ibcp.fr/ESPript/cgi-bin/ESPript.cgi). The structure domains of amino acid sequence were analyzed using the Conserved Domains Database (CDD, https://www.ncbi.nlm.nih.gov/Structure/cdd/wrpsb.cgi). A phylogenetic tree was calculated using the MEGA 7.0 software by neighbor-joining method and bootstrap replicates with 1000 times. The GenBank accession numbers of *AMH* sequences among various species used for sequence characterization, homology and phylogenetic analysis were listed in Appendix A.

The Putative Transcription Start Sites of the Goose AMH Gene were Predicted Using Neural Network Promoter Prediction (http://www.fruitfly.org/seq_tools/promoter.html). The Promoter Typical Components were Analyzed by the Plant Care (http://bioinformatics.psb.ugent.be/webtools/plantcare/html/). The Crucial Transcription Factor Binding Sites were Predicted by the TF SEARCH (http://diyhpl.us/~bryan/irc/protocol-online/protocol cache/TFSEARCH.html). Possible CpG Islands were Searched for Using DataBase of CpG Islands & Analytical Tools (http://dbcat.cgm.ntu.edu.tw/).

### 2.4. Quantitative Real-Time PCR

The expression profiles of *AMH* mRNA were measured in granulosa layers of different sized follicles by quantitative polymerase chain reaction (q-PCR). Q-PCR reactions were performed in a CFX96TM Real-Time System (Bio-Rad, California, USA) using SYBR PrimerScriptTM RT-PCR kit (Takara, Dalian, China). The procedures included 95 ℃ for 30 s, followed by 40 cycles of 95 °C for 5 s imposing the annealing temperature for 30 s and at 72 °C for 10 min. An 80-cycle melting curve was performed, starting at a temperature of 65 °C and increasing by 0.5 °C every 10 s to determine primer specificity. Each sample was in triplicate and the expression profile of *AMH* mRNA was evaluated by 2^−ΔΔCt^ method [26] by normalizing to *GAPDH* and *β-actin*. Primers designed for qPCR were listed in Appendix A.

### 2.5. Construction of the Geese AMH Promoter Luciferase Plasmid

The goose *AMH* promoter reporter vectors were constructed by PCR cloning. Seven specific primers containing *Kpn I* and *Xho I* restriction enzyme cleavage sites were designed to amplify the desired promoter fragments of the geese *AMH* gene and the PCR products were cloned into a *Kpn I*- and *Xho I*-digested pGL4.10-Basic vector (Promega, Wisconsin, USA). The primers including restriction enzyme sites were listed in Appendix A. According to the length of the vector, the plasmids were named pGL4.10-AMH1 to pGL4.10-AMH7. All the constructs were confirmed by dual-enzyme digestion and sequencing.

Mutation vectors of three binding sites for GATA4 (−778 bp, −1399 bp and −1477 bp) of the geese *AMH* promoter were generated using the Fast Site-Directed Mutagenesis Kit (Tiangen, Beijing, China). Mutagenic primers were listed in Appendix A. The G base at the binding site was mutated into C base where the promoter region of *AMH* could not be bound. These mutation vectors were named pGL4.10-AMH2-GATA4-778, 1399, 1477 (referred to as M778, M1399, M1477), respectively. All the plasmids were confirmed by DNA sequencing.

### 2.6. Cell Transfection and Luciferase Activity Analysis

Chinese Hamster Ovary (CHO) Cells Line was Obtained from Kunming Cell Bank, Chinese Academy of Sciences and was Incubated for Dual-Luciferase Reporter Assay. The Culture Medium Included High Glucose Dulbecco’s Modified Eagle’s Medium (DMEM, HyClone, China), 10% Fetal Bovine Serum (FBS, Gibco, USA) and 1% Streptomycin and Penicillin Mixture (Gibico, California, USA). Cells were Incubated at 37 °C in 5% CO_2_-Humidified Atmosphere to Reach 50–60% Confluence for Transfection. CHO Cells were Seeded into 96-well Plates at a Density of 1 × 10^4^/Well.

Subsequently, plasmids were transfected into CHO cells using Lipofectamine^®^ 3000 Transfection Reagent (Invitrogen, California, USA) according to the manufacture’s protocol. For dual-luciferase reporter assays, CHO cells were transfected with 100 ng/well of reporter gene plasmids (PGL4.10-AMH), pRL-TK renilla luciferase plasmids (as an internal control) and expression plasmids (Promega, California, USA) or pAc5.1-GFP plasmids (as control) in 96-well plate. After transfection for 24 h, the cells were collected. Luciferase activities were measured using the Dual-Luciferase^®^ Reporter Assay System (Promega, California, USA) and promoter activity was calculated as firefly-luciferase activity/sea pansy-luciferase activity. All transfections were performed in triplicate.

### 2.7. Statistical Analysis

All data were presented as the mean ± SD. Data were analyzed using the GraphPad Prism 5.0 Software (city, state abbrev. if USA, Country). The normal distribution and homogeneity of variances of all data were tested with a Shapiro-Wilk test [27] and Levene’s test [28], respectively. Multiple comparisons were performed using an ANOVA followed by the Turkey’s test using SPSS. 24.0. *p* < 0.05 was considered statistically significant.

## 3. Results

### 3.1. Sequence Analysis of CDS of Goose AMH Gene

After sequencing, the full-length coding sequence of goose *AMH* was obtained (Figure 1A). The complete coding sequence of goose *AMH* (GenBank accession No. MK061535) contained 2013 nucleotides encoding 670 amino acids. Homology analysis of goose *AMH* showed that there was a high sequence homology with other avian species such as duck (95.0%) and chicken (85.1%) (Appendix A). The conserved domains analysis indicated that the goose *AMH* amino acid contained two domain characteristics of the AMH family: AMH N-terminal region (AMH_N) at the N-terminus and transforming growth factor-β (TGF-β)-like domain (TGFB) at the C-terminus (Figure 1B). Homology alignment of *AMH* amino acid in various species showed that there were greater homologies in the TGFB domain (Appendix A). In addition, a phylogenetic tree was constructed based on the AMH amino acid sequences for geese and other species (Figure 2), which indicated that the goose *AMH* clustered with the clade of *Anas platyrhynchos* and *Gallus gallus*.

### 3.2. Sequence Analysis of 5′-Flanking Region of Goose AMH Gene

After sequencing, approximately 2.3 kb upstream sequence started from the translation start site (ATG) of the goose *AMH* was obtained (Appendix A). Analysis of sequence showed that five core transcription promoter regions were predicted in the 5′-flanking sequence of the *AMH* gene, including the regions of −2368 to −2319, −1763 to −1714, −538 to −489, −317 to −268 and −109 to −60, respectively (Table 1, Appendix A). Among these regions, the −109 to −60 was speculated to be the main core promoter region according to the score which was predicted by a software (Table 1). Furthermore, the core promoter elements including TATA-box, GATA-box were predicted in the 5′-flanking sequence of goose *AMH* gene (Figure 3A). Several transcription factors were also found in the 5′-flanking sequence of goose *AMH* gene, including neurofibromin 1 (NF-1), AP-1 binding site (AP-1), AP-2 binding site (AP-2), SP1, forkhead box M1 (FOXM1) and GATA-4 (Figure 3B). In addition, a CpG island was also detected at the region of −1127 to −1295 in the *AMH* promoter (Figure 3C). The length of the CpG island was 169 bp and the content of the GC was above 60%.

### 3.3. Expression Profiles of AMH mRNA in Granulosa Layers during Follicle Development

As depicted in Figure 4, the expression levels of *AMH* mRNA in granulosa layers gradually decreased during the development of follicles. The highest mRNA level of *AMH* was detected in the 4–8 mm follicles (Figure 4, *p* < 0.05). Besides, the mRNA levels of *AMH* in the 4–10 mm follicular granulosa layers were significantly higher than that in the F5–F1 follicles (Figure 4, *p* < 0.05) and there was no evident difference in the F5–F1 follicular granulosa layers (Figure 4, *p* > 0.05).

### 3.4. Promoter Activity Analysis of Goose AMH Gene

To characterize the promoter activities of the 5′-flanking sequence and identify the putative essential regulatory elements for transcription of the goose *AMH* gene, seven deletion promoter reporter vectors (pGL4.10-AMH 1 ~ 7) were constructed successfully (Appendix A) and the ATG was defined as +1. The promoter activities of goose *AMH* gene were analyzed in CHO cell lines and were shown in Figure 5. The result showed that deletion of this distal promoter fragment from −2344 until −637 bp did not increase the promoter activity significantly (Figure 5, *p* > 0.05). However, further deletion of −331 to −1 bp significantly increased the promoter activity (Figure 5, *p* < 0.05), while deletion of −87 to −1 bp significantly decreased the promoter activity (Figure 5, *p* < 0.05), suggesting that there was a negative regulatory element in the region within −637 to −331 bp and a positive regulatory element in the region within −331 to −87 bp.

### 3.5. Regulatory of GATA4 on AMH Gene Promoter Activity

As depicted in Figure 3B, GATA-4 was found in the 5′-flanking sequence of goose *AMH* gene. Three putative binding sites of GATA-4 were predicted at −778 bp, −1399 bp and −1477 bp, respectively (Appendix A). After that, the luciferase assay was performed in CHO cells. Forty-eight hours after culture, the CHO cells became almost confluent in a culture dish (Appendix A) and green fluorescence appeared in the nuclei of CHO cells (Appendix A). The expression level of *GATA-4* mRNA was significantly higher than that of control and pEGFP-N1 group (an internal control) (Figure 6A, *p* < 0.05), which excluded the possibility of endogenous expression of *GATA-4* in CHO cells. To analyze whether GATA-4 could regulate the transcription of *AMH*, a reporter gene assay was developed in CHO cells by overexpressing *GATA-4*. Results showed that the relative luciferase activity significantly increased in co-expression of *AMH* and *GATA-4* (Figure 6B, *p* < 0.05). When the putative binding sites of GATA-4 were mutated in the 5′-flanking sequence of *AMH*, the relative luciferase activity significantly decreased in both −778 and −1477 bp binding sites (Figure 6B, *p* < 0.05). The luciferase activity was the lowest in −1477 bp binding site (Figure 6B, *p* < 0.05), indicating the importance of the −1477 bp binding site of *GATA-4* for the transcription of *AMH* promoter.

## 4. Discussion

*AMH* is an important factor regulating sex differentiation and is also produced by GCs from the ovaries [7,29,30]. To date, the *AMH* gene has been cloned in various species such as mouse, rat, pig and chicken [31,32,33,34]. However, neither its full-length CDS sequence nor its function in the ovary has been reported in geese. Herein, our results showed that the complete CDS of geese *AMH* gene had a higher homology with that of chicken and duck and the phylogenetic analysis indicated that goose *AMH* was clustered with chicken and duck into the same clade. These data suggested that *AMH* gene might be highly conserved among goose, chicken and duck. Further analysis of the amino acid sequence showed that goose AMH contained an AMH-N domain at the N terminus and a TGF-β domain at the C terminus. The TGF-β domain shared a high homology by alignment with the counterparts of other species, which was consistent with the result of a previous study where Hu et al. showed that a high homology of the *AMH* amino acid at the C terminus was due to the conservatism of the TGF-β family among different species [35]. While the N terminus of chicken *AMH* amino acid was longer and diverged from the counterpart in mammals, the divergence resulted in low homology [34]. Despite that, the main features of the AMH protein were conserved between chicken and mammals [36]. Therefore, these results suggested that the functions of goose *AMH* might be similar to other species and the TGF-β domain might be a crucial region for its functions.

We also found that the relative expression of goose *AMH* decreased gradually with the increasing follicular diameter, which was in line with the previous studies in chicken [15,36]. In chicken, the expression level of *AMH* was related to the growth of small follicles [37]. Several evidence has also shown that *AMH* inhibited the activation of primordial follicles [38], attenuated the rate of follicular growth [39] and inhibited follicular recruitment [40], suggesting the inhibitory action of *AMH* in the selection and growth of small follicles. It was also found that the expression level of *AMH* in GCs of the 4–10 mm follicles was higher than that in those of the F5–F1 follicles, suggesting that *AMH* might play a negative role in geese follicle recruitment and selection. Recently, *AMH* has been suggested to be a prognostic biomarker for small yellow follicles (SYFs) and identified as a reliable marker of ovarian reserve [41,42], indicating the importance of *AMH* during the early development of follicles.

In order to reveal the transcriptional regulatory mechanism of goose *AMH*, its 5′-flanking sequence was cloned in the present study. Subsequent analysis predicted several core promoter elements in its 5′-flanking sequence as well as potential transcription factors. The promoters of rodent and human *AMH* shared common transcription factors that were tightly clustered in their 5′ proximal sequences, including the SOX9, NR5A1 and GATA-4 (reviewed in References [43,44]). Result of Yuan et al. showed that the relative expression of goose *GATA-4* gradually decreased during the development of follicles [45], which was consistent with expression pattern of *AMH*. In addition, the luciferase activity significantly increased when co-expressing *AMH* and *GATA-4*, implicating GATA-4 as a positive regulatory factor for *AMH* gene expression. A previous study demonstrated that the lack of *GATA-4* expression within the ovary caused a failure during the development of follicles [46]. The roles of GATA factors in gonadogenesis have been further elucidated by identifying their downstream target genes, which including *AMH* [47,48,49]. Our results showed that there were essential regulatory elements in the region within −637 to −87 bp which was located in the proximal region of *AMH* promoter and might be the crucial region to activate *AMH* transcription. Therefore, the binding sites of GATA-4 to *AMH* were speculated to locate in the proximal region of *AMH* promoter. However, the predicted binding sites of GATA-4 in the promoter region of *AMH* were located at −778 bp, −1399 bp and −1477 bp, respectively. The dual-luciferase activity significantly decreased when the binding sites of GATA-4 at −778 bp and −1477 bp were mutated. These results implied that GATA-4 might regulate *AMH* transcription by binding to its distal promoter regions. Furthermore, the luciferase activity significantly decreased at the −1477 bp binding site than −778 bp. Previous studies showed that GATA-4 binding to the *AMH* promoter was essential to maintain *AMH* transcription [50] but GATA-4 by itself was a poor activator of the *AMH* promoter [51]. Studies showed that *AMH* gene expression was regulated by transcriptionally cooperation of two transcription factors, such as FOLX2 and SF-1 [23], GATAs and SF-1 [48]. Therefore, it is possible to speculate that there might be another factors to synergistically enhance the action of GATA-4 at the binding site of −778 bp, which was needed to be verified further.

## 5. Conclusions

In conclusion, the intact CDS of goose *AMH* gene was obtained, which encoded for 670 amino acids consisting of an AMH-N domain at the N terminus and a TGF-β family at the C terminus. The promoter sequence of *AMH* contained five core transcription regions and there were essential regulatory elements spanning from the −637 to −87 bp. The relative expression of *AMH* in granulosa layers gradually decreased during the development of follicles. The luciferase activity significantly increased in co-expression of *AMH* and *GATA-4*. However, when the binding sites of GATA-4 at the −778 bp and −1477 bp were mutated respectively, the luciferase activities significantly decreased. These results suggested that the transcriptional activity of *AMH* was activated by GATA-4 to inhibit the development of small follicles in goose.

## Figures and Tables

**Figure 1 animals-09-00816-f001:**
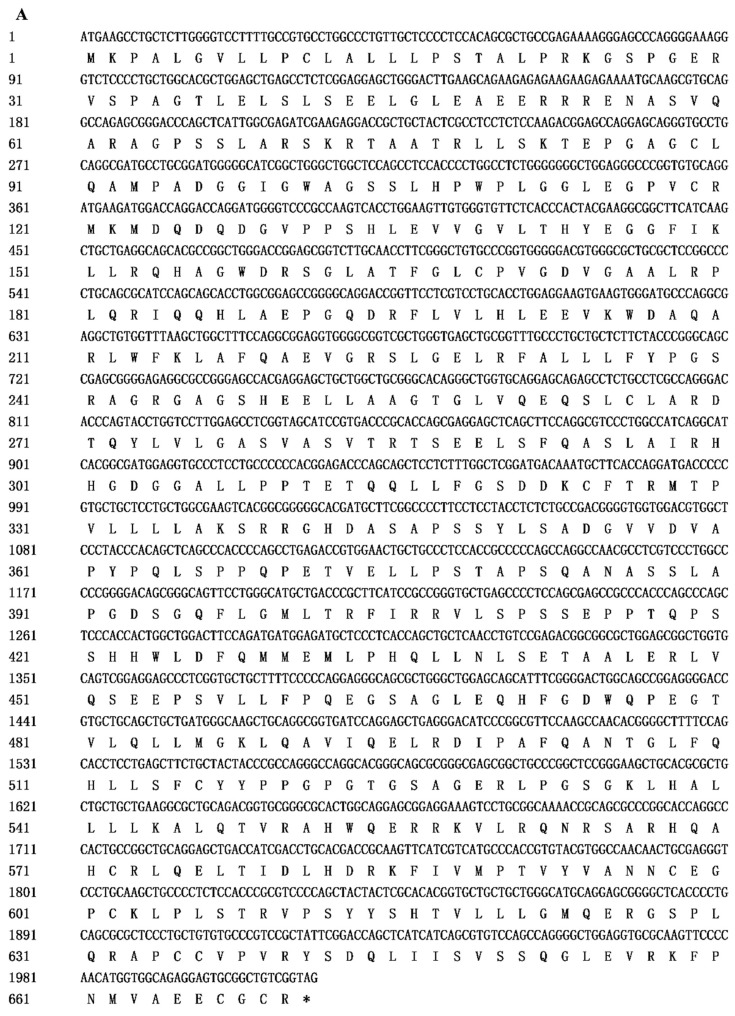
Nucleotide and deduced amino acid sequences as well as the predicted structural domains of anti-Müllerian hormone (*AMH*) gene in geese. (**A**) Nucleotides and amino acids sequence were displayed from the 5′ to the 3′ direction and the locations of both nucleotide and amino acid were numbered on the left side. The entire deduced amino acid sequence is depicted in single letter code beneath the corresponding nucleotide sequence. The stop codon TGA is with ‘*’. (**B**) The location of predicted structural domains in the amino acid sequence of goose *AMH*. The number upon the query sequence represents the number of amino acids.

**Figure 2 animals-09-00816-f002:**
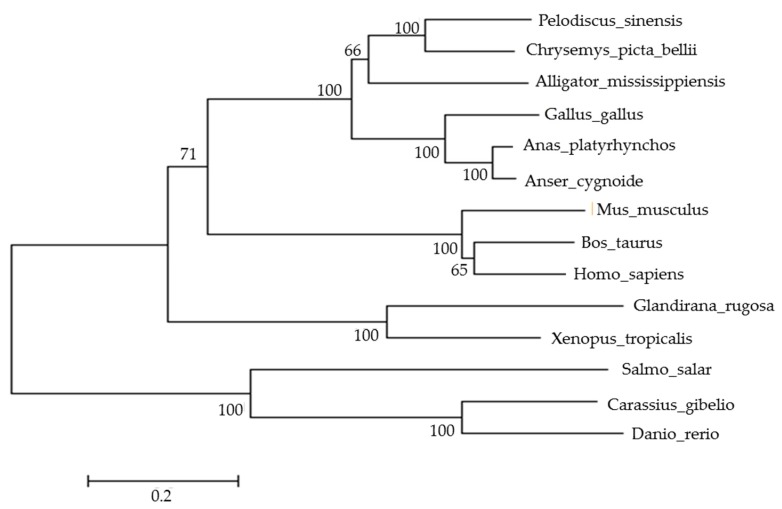
A phylogenetic tree of AMH amino acid sequence among various species. The *AMH* sequences from other species were constructed by using the maximum likelihood method. The bootstrap sampling was performed with 1000 replicates and the numbers at the forks indicate the bootstrap proportions. The scale bar represents the expected number of amino acid substitutions per site.

**Figure 3 animals-09-00816-f003:**
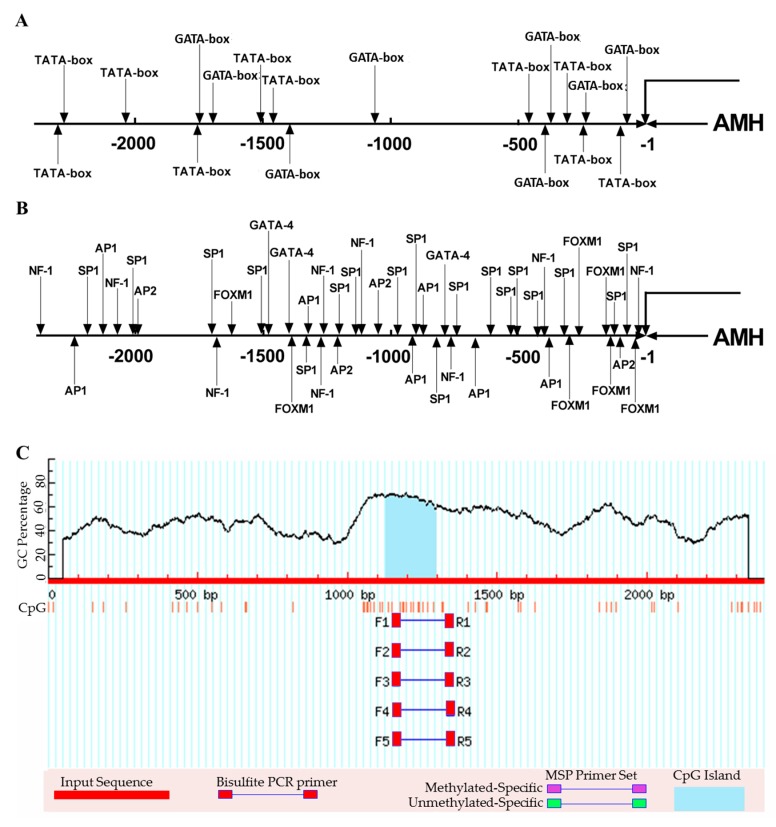
Analysis of the promoter region of *AMH* gene in geese. (**A**) The prediction of core promoter elements in 5′-flanking region of goose *AMH* gene. (**B**) The prediction of transcription factors in 5′-flanking region of goose *AMH* gene. The line represented the *AMH* promoter region and ‘−1′ represented the translation initial site ‘ATG.’ The number below the line represented the positions of promoter sequence. The arrows represented the positions of core promoter elements and transcription factors, respectively. (**C**) The prediction of CpG island of goose *AMH*. There was a CpG island being predicted.

**Figure 4 animals-09-00816-f004:**
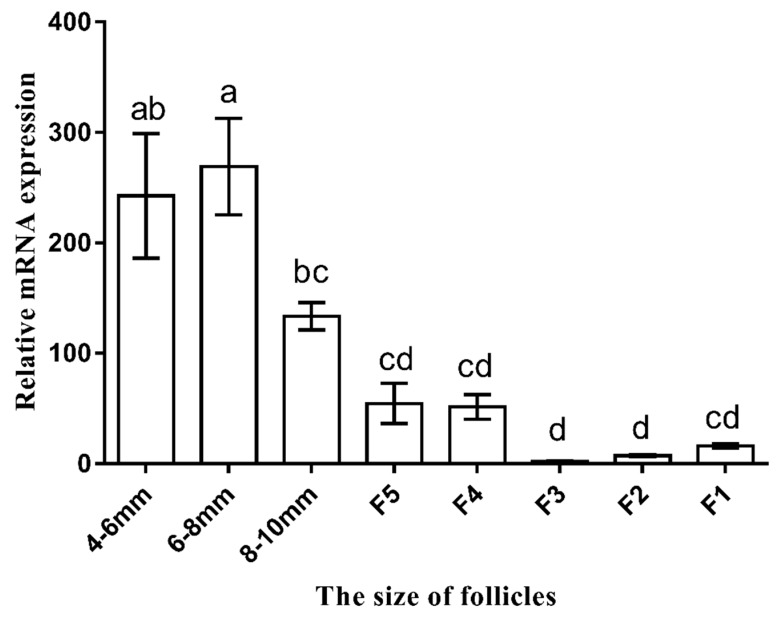
Expression profile of *AMH* mRNA in granulosa layers of follicles at different development stages in geese. The values for expression profile were normalized to *GAPDH* and *β-actin* and were compared using the 2^−ΔΔCt^ method. The data were presented as the mean ± SD (*n* = 3). Different lowercase letters represented the evident difference in granulosa cells among stages of follicular development (*p* < 0.05). F5 represents the fifth largest follicle, F4 represents the fourth largest follicle, F3 represents the third largest follicle, F2 represents the second largest follicle and F1 represents the first largest follicle.

**Figure 5 animals-09-00816-f005:**
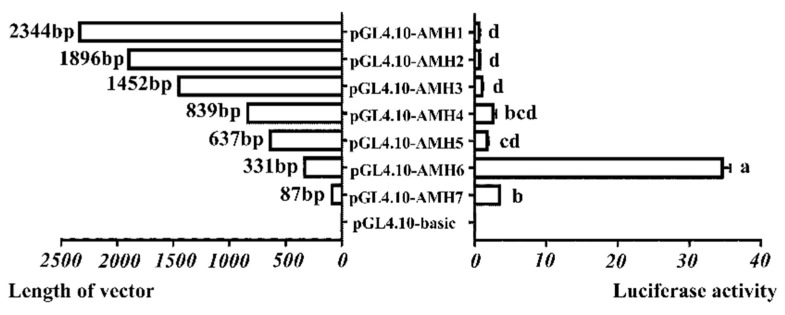
Analysis of *AMH* gene promoter activity in geese. Luciferase report vectors containing various lengths of the goose *AMH* promoter region were constructed and transfected into Chinese hamster ovary (CHO) cells. Various lengths of goose *AMH* region contained pGL4.10-AMH1, deletion of −2344 to −1 bp; pGL4.10-AMH2, deletion of −1896 to −1 bp; pGL4.10-AMH3, deletion of −1452 to −1 bp; pGL4.10-AMH4, deletion of −839 to −1 bp; pGL4.10-AMH5, deletion of −637 to −1 bp; pGL4.10-AMH6, deletion of −331 to −1 bp and pGL4.10-AMH7, deletion of −87 to −1 bp. The left bars indicated truncated *AMH* promoter sequences linked to luciferase, the bars on the right indicated luciferase activity relative to the pGL4.10-basic negative control vector. Each column represented the mean ± SD (*n* = 3). Different lowercase letters represented the evident difference (*p* < 0.05).

**Figure 6 animals-09-00816-f006:**
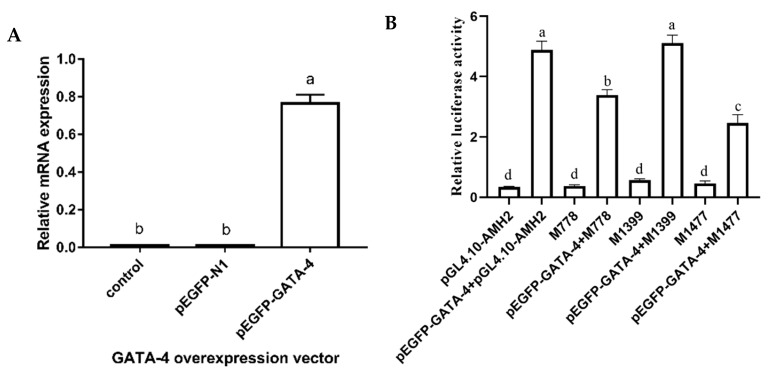
The effect of GATA-4 binding sites on the transcription regulation of *AMH* gene in geese. (**A**) Overexpression of GATA-4 in the CHO cells. pEGFP-N1 represented an internal control. pEGFP-GATA-4 represented the overexpression plasmid of *GATA-4* gene. (**B**) Three binding sites of GATA-4 in the promoter region of goose *AMH* and mutation constructions were transfected into CHO cells. pGL4.10–AMH2 represented the wild binding sites of GATA-4 in the promoter region of goose *AMH*. M778, M1399 and M1477 represented the mutation binding sites of GATA-4 in the promoter region of goose *AMH*, respectively. pEGFP-GATA-4 + pGL4.10-AMH2, pEGFP-GATA-4 + M778, pEGFP-GATA-4 + M1399 and pEGFP-GATA-4 + M1477 represented the co-expression transfection of pEGFP-GATA-4 and three mutation binding sites in the CHO cells, respectively. Each column represented the mean ± SD (*n* = 3). Different lowercase letters represented the evident difference (*p* < 0.05).

**Table 1 animals-09-00816-t001:** The prediction results of *AMH* gene core transcriptional promoter region in geese.

Number	Start	End	Score	Promoter Sequence (5′ to 3′)
1	−2368	−2319	0.99	GTGTATTGGTTAAAAATGTTGCTGTGTGTTCAGTTTCTTT **A**GCCCTTAAA
2	−1763	−1714	0.92	ATTGATGATTTAAAAAAAGACAATCATCCTTCCCCGAAGC **G**CAGCTGCCT
3	−538	−489	0.90	TTAAGAGCCTCTGTAAGGCGGCCTCTACACCACGGTGCAC **A**GGGTGCCAG
4	−317	−268	0.95	TCCACCCTCCTTTAAAAAACATCTGAAGTCAAGGACGTGC **A**TCCCTACAA
5	−109	−60	1	GCCACCGGTTTTTAAAAGGGAGATGCCGTCCTCCCCTTCC **G**CGCAAATGT

The core transcription region has an evaluation score of 0–1 points and a minimum threshold of 0.8 points. The promoter sequence plus bases are the predicted transcription start site of the goose *AMH* gene.

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
