# Peer review of "Promoter Identification and Transcriptional Regulation of the Goose AMH Gene"

_animals, 2019, doi:10.3390/ani9100816_

Round 1

Reviewer 1 Report

In this study, sequences of the coding region of goose AMH gene and the 5’ proximal sequence regarded as the transcription regulation region were identified. In addition, the ability of transcription regulation of the identified 5’ proximal region was partially determined. Because AMH gene is known to be involved in the control of follicular development in many animals including humans, the results of this study has certain value even though the authors focused on goose. The strategy and the methods in this manuscript are appropriate and the conclusion is reasonable. The amount and the quality of the data are enough to support the objective. However, several points are difficult to understand and should be revised properly.

Line 59: “Promoter is located in the upstream of one gene and plays a pivotal role in gene expression.” should be “Promoter is usually (commonly) located in the 5’ proximal of one gene…”. Promoter is usually found in the 5’ proximal region of the coding region. It is a rare case but there are genes do not have the promoters in the 5’ proximity of the coding region.

Line 63: “PCOS” is an abbreviation. As this word is used once in the manuscript, this should be just stated as “polycystic ovary syndrome”.

Line 73: The merit to reveal the sequence of AMH gene and the transcription regulation in geese should be added around here. This reviewer guesses the classification of the developmental stage of the sampled follicles in the ovary was usually easier in avians than mammals. In this regard, avians may be better model to analyze the transition of the level of gene expression in the follicles along with the development.

Line 88-89: The classification of the follicles are difficult to understand. Do the statements about the size “4-6mm…” indicate diameter of the follicles? Moreover, this reviewer has completely no idea what "F1-F5" mean. It seems that F1 is the biggest follicles and the size gets smaller toward F5. However, what did the authors measure for the classification of F1 to F5? This cannot be the diameter, because the classification by the diameter has already indicated as “4-6 mm, 6-8 mm and 10 mm”. In addition, how did the authors distinguish the growing follicles from the degrading follicles?

Line 95-96: The strategy to prepare the primers to isolate the CDS and 5’proximal region of CDS is not stated fully. As far as read the lines 95-96, NW_013185857.1 seems to be the sequence used for the preparation of the primers. However, this sequence is a result of the shotgun sequence of goose DNA and there are no indications that this sequence includes the sequence of AMH gene in the website of NW_013185857.1. How did the authors choose NW_013185857.1 as a candidate sequence including AMH gene? This information should be stated properly in the text to make sure the obtained sequences in Fig. 1 is really the sequence of AMH gene of goose (This reviewer believes that the sequence shown in Fig.1 is AMH gene of goose. It is obvious from Fig. S1. However, the methods should be stated clearly.).

Line 108: How do we believe that the primers shown in Table 1 can detect the full-length of CDS of AMH gene of goose? This question relates to the question given for line 95-96.

Line 108: This line should be “Primers used for cloning full-length CDS of AMH gene in geese”. The tables and figures should be prepared so that the readers understand what was done in the experiments without reading the text in the paper. The tables and figures should be revised thoroughly in this regard.

Line 114: Should “AMH” be italicized? There are many of same mistakes in this manuscript. The authors should be revised all of them.

Line 124-129:

Is there an explanation for how to detect the elements (TATA-box, CATA-box) in the text?

The authors showed the mutations put in the GATA-4 binding site (Does GATA-4 binding site mean GATA box?) in Table 6. This reviewer is not convinced that the inserted mutations can affect the binding ability of GATA-4 to GATA-4 binding site (GATA-box). For example, the effect of M1339 was less than those of the other mutations (Fig.6B). Is it just because M1339 did not affect the binding ability of GATA-4 to the GATA-4 binding site (GATA box) and the other mutations reduced the binding ability?

To avoid such criticism, the consensus sequence of GATA-4 binding site/GATA-box should be stated and compared with the mutated sequences shown in Table 6.

Line 196: The supplemental figures stated multiple ways like “supplement1” or “Sup. 1”. Lines 222 and 254 should be also checked.

Lines 222-224: How did the authors judge that the -2328 to -2278 is the predicted core promoter? Is it because this region include many elements such as TATA-box and CATA-box? The authors seemed to use a software to determine this region as a core promoter and may be difficult to explain how to choose this region as a core promoter. However, to state just “-2328 to -2278 is judged to be the predicted core promoter” in here is too abrupt and some criterion should be stated concisely.

Line 225: Does “enriched” a correct expression? Just “found” is good enough?

Line 231: The explanation should be given for Fig.3A and B separately. It seems that A is for the distribution of the elements and B is for the distribution of the transcription binding sites.

Lines 242-244: This reviewer has no idea which data the information stated here is derived from. Did it relate to the comparison of F1 to F5 in Fig. 4?

Lines 246-250: In the figure legend, the explanations of F1 to 5 should be added. All abbreviations in the figure and tables should be clearly explained as figure legend or footnotes. All figures and tables in this manuscript should be checked in this regard.

Line 260: In Fig. 5, it seems that the deletion of -331 to -87 bp affected the transcription promotion activity of the promoter region. This should be mentioned in result section and discussion section.

Line 277 and Fig. 6: Is there an explanation about what pEGFP-N and pEGFP-GATA?

Lines 309-310: “While….regions.” is difficult to understand.

Lines 319-320: It is difficult to understand which figure the statement in these lines based on. The comment in here relates to the comment given for lines 242-244.

Lines 330-332: The expression of GATA-4 should be examined in the sampled ovarian follicles to support the authors’ idea that the transcriptional activity of AMH gene was activated by GATA-4 to inhibit the development of small follicles.

Other comments

Tables: Several tables including the primers used for cloning, making vectors and q-PCR can be put into supplemental files.

Figures: The quality of the figures is bad. Especially, Figure 3 is awful.

English in this manuscript should be corrected by native speaker.

Reviewer 2 Report

The review of the manuscript titled: “Promoter identification and transcriptional regulation of the goose AMH gene” by Yang et al. submitted to Animals journal

The manuscript concern the description and characterisation of the coding sequence and regulatory elements in AMH gene in goose. The manuscript is interesting and brings new information concerning studied gene, however there are some mistakes or discrepancies in the test which must be corrected before publication. Detailed comments for the author which should be considered are listed below:

1) ….a cis regulatory element spanning from -637 to -331 bp was identified. – this location should be indicated in the Figure S2

2) …encoding 403 amino acids – should be 670 amino acids

3) the tables with primer’s sequences (Table 1, 2, 5, 6) should be moved to supplementary material

4) line 190 – according to GenBank: MK061535 the AHM gene encodes a polypeptide of 670 amino acids (since at the position p.671 there is a stop codon)

5) line 196: (supplement 1) – should be Figure S1

6) Figures 1A, Figure 3, as well as Figures S1, S2, S3 – are blurred – very bad quality (not focus) and should be corrected

7) The title of Figure 2 should be: A phylogenetic tree of AMH amino acid sequence among different species.

8) Figure 2 – the name for goose should be changed for latin since for all species the latin name is given

9) line 219 and 222: supplement 2 and Sup 2 – should be Figure S2

10) line 221: ….including the regions of -69 to -19, -674 to -624, -1899 to -1849, -2120 to -2070 and -2328 to -2278 – the numbering is incorrect. The -1 nucleotide is the first nucleotide before adenine from ATG codon and not the first from the beginning of the sequence presented – all numbering must be corrected

11) line 222: ….the -2328 to -2278 was speculated….. – according to comment above, this region should be numbered as -60 to -104

12) line 223: the TATA-box and CATA-box – should be marked in Figure S2

13) Table 7 – the start and end positions should be corrected according to comment no. 10

14) line 242 - …4 - 8 mm follicles... – should be 4-10 mm follicles

15) line 254: (Sup. 3) – should be Figure S3

16) line 275: (Sup 4A) – should be Figure S4A

17) line 276: (Sup 4B) – should be Figure S4B

18) Figure S1 – very poor quality and in the legend (The name of species were numbered on the left side.) – no numbering for the species is applied

19) Figure S2 – numbering on the right site should start from -1 before ATG and reached -2389 at the top of the figure!
